# Identifying the Posture of Young Adults in Walking Videos by Using a Fusion Artificial Intelligent Method

**DOI:** 10.3390/bios12050295

**Published:** 2022-05-03

**Authors:** Posen Lee, Tai-Been Chen, Chin-Hsuan Liu, Chi-Yuan Wang, Guan-Hua Huang, Nan-Han Lu

**Affiliations:** 1Department of Occupation Therapy, I-Shou University, No. 8, Yida Road, Jiaosu Village, Yanchao District, Kaohsiung 82445, Taiwan; posenlee@isu.edu.tw; 2Department of Medical Imaging and Radiological Science, I-Shou University, No. 8, Yida Road, Jiaosu Village Yanchao District, Kaohsiung 82445, Taiwan; ctb@isu.edu.tw (T.-B.C.); wang1b011@isu.edu.tw (C.-Y.W.); ed103911@edah.org.tw (N.-H.L.); 3Institute of Statistics, National Yang Ming Chiao Tung University, No. 1001, University Road, Hsinchu 30010, Taiwan; ghuang@stat.nctu.edu.tw; 4Department of Occupational Therapy, Kaohsiung Municipal Kai-Syuan Psychiatric Hospital, No. 130, Kaisyuan 2nd Road, Lingya District, Kaohsiung 80276, Taiwan; 5Department of Pharmacy, Tajen University, No. 20, Weixin Road, Yanpu Township, Pingtung County 90741, Taiwan; 6Department of Radiology, E-DA Hospital, I-Shou University, No. 1, Yida Road, Jiaosu Village, Yanchao District, Kaohsiung City 82445, Taiwan

**Keywords:** iso-block postural identity, OpenPose, fusion deep learning

## Abstract

Many neurological and musculoskeletal disorders are associated with problems related to postural movement. Noninvasive tracking devices are used to record, analyze, measure, and detect the postural control of the body, which may indicate health problems in real time. A total of 35 young adults without any health problems were recruited for this study to participate in a walking experiment. An iso-block postural identity method was used to quantitatively analyze posture control and walking behavior. The participants who exhibited straightforward walking and skewed walking were defined as the control and experimental groups, respectively. Fusion deep learning was applied to generate dynamic joint node plots by using OpenPose-based methods, and skewness was qualitatively analyzed using convolutional neural networks. The maximum specificity and sensitivity achieved using a combination of ResNet101 and the naïve Bayes classifier were 0.84 and 0.87, respectively. The proposed approach successfully combines cell phone camera recordings, cloud storage, and fusion deep learning for posture estimation and classification.

## 1. Introduction

The OpenPose algorithm is a deep learning method in which part affinity fields (PAFs) are used to detect the two-dimensional (2D) postures of humans in images [1]. The relationship between posture stability, motor function, and quality of life has been determined [2,3]. Moreover, the OpenPose algorithm has been used for checking the medication situations of patients and for their physical monitoring [4,5]. The evaluation of the cardinal symptoms of resting tremor and bradykinesia for Parkinson's disease has been conducted using an OpenPose-based deep learning method [6,7]. Furthermore, in [8], the OpenPose framework was used to create a human behavior recognition system for skeleton posture estimation. Quantitative gait (motor) variables can be estimated and recorded using pose tracking systems (e.g., OpenPose, AlphaPose, and Detectron) [9]. These factors are useful for measuring the quality of life of older adults [10,11,12]. Moreover, parkinsonian motion features have been created using deep-learning-based 2D OpenPose models [13,14]. For people with autism spectrum disorder, skeleton posture characteristics are correlated with long-term memory in the field of action recognition [15,16,17]. The physical function of a patient should be assessed according to their health data obtained using a skeleton pose tracking device and gait analysis [18,19,20,21]. Many neurological and musculoskeletal disorders are associated with problems related to postural movement, which can be estimated using a pose-capturing device [22]. Therefore, noninvasive tracking devices are used to record, analyze, measure, and detect the postural control of the body, which may indicate health problems in real time. In this study, fusion deep learning was used to generate dynamic joint node plots (DJNPs) by using OpenPose-based methods, and skewness in walking was qualitatively analyzed using convolutional neural networks (CNNs) [23]. An iso-block postural identity (IPI) method was used to perform the quantified analysis of postural control and walking behavior. This proposed approach combines cell phone camera recordings, cloud storage, and fusion deep learning for postural estimation and classification.

## 2. Materials and Methods

### 2.1. Research Ethics

All the experimental procedures were approved by the Institutional Review Board of E-DA Hospital [with approval number EMRP52110N (04/11/2021)]. Verbal and written information on all the experimental details was provided to all the participants before they provided informed consent. Written informed consent was obtained from the participants prior to experimental data collection.

### 2.2. Flow of Research

In this study, videos walking toward and away from a cell phone camera were recorded using the camera (Step 1 in Figure 1). The videos were recorded at 24-bit (RGB), 1080p resolution, and 30 frames per second. The videos were uploaded to Google Cloud through 5G mobile Internet or Wi-Fi (Step 2 in Figure 1). The workstation used in this study downloaded a video, extracted a single frame from the video, and then applied a fusion artificial intelligence (AI) method to this frame (Step 3 in Figure 1). In the aforementioned step, single frames were extracted from an input video (Step 3A), frames with static walking were identified using an OpenPose-based deep learning method (Step 3B), and the joint nodes of the input video were merged into a plot (Step 3C). The obtained DJNP was categorized as representing straight or skewed walking (Step 3D). CNNs were used to classify DJNPs into one of the aforementioned two groups. Two types of deep learning methods were used in the fusion AI method adopted in this study: an OpenPose-based deep learning method and CNN-based methods. The OpenPose-based method is useful for estimating the coordinates of joint nodes from an input image [1]. The adopted CNNs are suitable for the classification of images with high accuracy and robustness.

### 2.3. Participants 

A total of 35 young adults without any health problems were recruited to participate in a walking experiment. The age range was 20.20 ± 1.08 years. The inclusion criteria were healthy adults who were willing to participate and could walk more than 5 m. People with musculoskeletal pain (such as muscle soreness), those who had drunk alcohol or taken sleeping pills within 24 h before the commencement of the experiment, and individuals with limited vision (such as nearsighted people without glasses) were excluded from this study.

### 2.4. Experimental Design

The experimental setup is depicted in Figure 2. The total length of the experimental space was greater than 7 m. The ground was level, free of debris, and smooth to ensure a straight and smooth walking path. The cell phone was placed 1 m above the ground (approximately equal to the height of a medium-sized adult holding a cell phone) and 2 m from the endpoint of the walking path. The entire body of a participant was recorded during the walk. The participants were required to wear walking shoes and not slippers while walking. Participants walked away from the cell phone and then turned back and walked toward the cell phone. The participants walked for 5 m toward and away from the camera three times each. One video was captured for each 5-m walk; thus, six videos were recorded for each participant. A series of single (static) frames was extracted from a video every 0.3 s. For example, for a 3-s input video, 10 frames were extracted to estimate the coordinates of joint nodes. A static frame of one DJNP was extracted per 0.3 s for one video. For example, a 10 s walking video with frame rate 30 (frames/second), the total static frame in one DJNP are 90 frames (i.e., 90 = 10 (second) × 30 (frames/second) × 0.3 (second)). Hence, the DJNP was a variety of frames according to the length of a walking video. The filmmakers are not medical experts but are trained in motion assessment. The video is analyzed by an expert in image analysis and an occupational therapist specializing in rehabilitation Table 1 lists the number of participants and the mean and standard deviation (STD) of velocity (m/s) and time (s) for each group.

### 2.5. Measurement of Joint Nodes through Openpose-Based Deep Learning 

OpenPose is a well-known system that uses a bottom-up approach for real-time multiperson body pose estimation. In the proposed OpenPose-based method, PAFs are used to obtain a nonparametric representation for associating body parts with individuals in an image [1]. This bottom-up method achieves high accuracy in real time, regardless of the number of people in the image. It can be used to detect the 2D poses of multiple people in an image and to perform single-person pose estimation for each detection. In this study, the OpenPose algorithm was mainly used to output a heat map of joint nodes (Figure 3). The center coordinates of joint nodes were estimated by using the geometric centroid formula.

### 2.6. Definition of the Control and Experimental Groups

The data for the control group comprised DJNPs that indicated straightforward walking toward and away from the camera. The experimental group comprised DJNPs that indicated skewed walking. The data for the control and experimental groups comprised 102 and 108 DJNPs, respectively, which were classified using different CNNs.

### 2.7. Classification Using Pretrained CNNs and Machine Learning Classifiers

Pretrained CNNs were used to extract the features of DJNPs, and machine learning classifiers were used to construct classification models. The eight pre-trained CNNs used in this study were AlexNet, DenseNet201, GoogleNet, MobileNetV2, ResNet101, ResNet50, VGG16, and VGG19. Moreover, the three machine learning classifiers used in this study were logistic regression (LR), naïve Bayes (NB), and support vector machine (SVM).

CNNs have a high learning capacity, which makes them suitable for image classification. They extract features and learn data according to variations in the breadth and depth of features. Table 2 lists the features that were extracted by CNNs and served as the inputs for the LR, NB, and SVM. A deep CNN network comprises five types of primary layers: a convolutional layer, a pooling layer, a rectified linear unit layer, fully connected layers, and a softmax layer. Information on the pretrained CNNs used in this study is provided in Table 2. The fully connected layers of the CNNs extracted and stored the features of the input image. In the present study, eight CNNs and three classifiers with four batch sizes and 20 random splits were adopted. The four batch sizes selected in this study for the CNNs were 5, 8, 11, and 14. The total number of investigated models was 8 (CNNs) × 3 (machine learning techniques) × 4 (batch size settings) × 20 (instances of random splitting) = 1920. Therefore, the 1920 models represent the 1920 possible combinations of one CNN, classifier, batch size, and random data split. CNNs have demonstrated utility and efficiency in image feature extraction in the fields of biomedicine and biology [23,24,25,26,27].

LR is a process of modeling the probability of a discrete outcome when an input variable is given. This process is often used to analyze associations between two or more predictors or variables. LR does not require the existence of a linear relationship between inputs and output variables. This method is useful when the response variable is binary, but the explanatory variables are continuous. LR is also an effective analysis method for classification problems. The LR method is used for the development of classification models in the field of machine learning because of its capacity to provide hierarchical or tree-like structures. Many fields have adopted LR for prediction and classification. LR is suitable for classification problems related to health issues, such as whether a person has a specific ailment or disease when a set of symptoms are given.

NB classifiers are based on Bayes’ theorem with a naïve independence hypothesis between the adopted predictors or features. These classifiers are the most suitable ones for solving classification problems in which no dependency exists between a particular feature and other features of a certain class. NB classifiers offer high flexibility for linear or nonlinear relations among variables (features or predictors) in classification problems and provide increased accuracy when combined with kernel density estimation. NB classifiers exhibit higher performance for categorical input data than for numerical input data. These classifiers are easy to implement and computationally inexpensive, perform well on large datasets with high dimensionality, and are extremely sensitive to feature selection.

SVM classifiers are highly powerful classifiers that can be used to solve two-class pattern recognition problems. They transform the original nonlinear data into a higher-dimensional space and then create a separating hyperplane defined by various support vectors in this space to maximize the margin between two datasets. Data can be linearly separated in the higher-dimensional space by using a kernel function. Many useful kernels are available to improve the classification performance and reduce the false rate. SVM is a supervised learning method for the classification of linear and nonlinear data and is generally used for the classification of high-dimensional or nonlinear data.

The computing time of using SVM is in linear time, rather than by expensive iterative approximation, which is performed by many other types of classifiers. The LR, NB, and SVM methods were applied as deep and machine learning methods to extract features of DJNPs and classify the postural control of the straight and skewed walking groups.

### 2.8. Validation of Classification Performance

The data for the control and experimental groups comprised 102 and 108 DJNPs, respectively. A random splitting schema was employed to separate the training (70%) and testing (30%) sets; 71 and 31 samples from the control group were used for training and testing, respectively, and 76 and 32 samples from the experimental group were used for training and testing, respectively. Testing sets and confusion matrices were used to evaluate the models with respect to the kappa value, accuracy, sensitivity, specificity, positive predictive value (PPV), and negative predictive value (NPV). These indices were sorted in the ascending order of the corresponding kappa value, and a radar plot was then generated to present the aforementioned indices of the adopted models.

## 3. Results

In this study, 70% of the samples of each group were randomly used to train the adopted classifiers, and the remaining 30% of samples were used to perform validation. Figure 4 shows a scatter plot for the specificity and sensitivity of the 1920 models for the validation dataset. The maximum specificity and sensitivity of 0.84 and 0.87, respectively, were achieved by the ResNet101 and NB classifiers, respectively.

In Figure 5, a radar plot was constructed for six performance indices with the results sorted by the maximum kappa value for 96 models (the abbreviations of the investigated models are written in Appendix A). The best performing model was M53, which is a combination of ResNet101 and naïve Bayes. The kappa, accuracy, sensitivity (Sen), specificity (Spe), PPV, and NPV values were 0.71, 0.86, 0.87, 0.84, 0.84, and 0.87, respectively. All of the performance indices are over 0.7. The optimized model, ResNet101 with naïve Bayes, had acceptable agreement results and the highest accuracy.

Table 3 lists the 13 models with kappa values greater than 0.59. These models comprised four (30.8%) AlexNet models, three DenseNet201 models (23.1%), three ResNet101 models (23.1%), two VGG16 models (15.4%), and one VGG19 model (7.7%). AlexNet, DenseNet201, and ResNet101 accounted for 10 of the aforementioned 13 models (76.9%). SVM and NB were the main machine learning classifiers that performed well in this study. The numbers of the aforementioned 13 models with SVM and NB classifiers were 3 and 10, respectively. Thus, NB performed well. Finally, the batch sizes of the 13 models were 5, 8, 11, and 14 useable in this work.

## 4. Discussion

### 4.1. Measurement of Postural Control

IPIs were used to measure the skewness or displacement. Figure 6 illustrates the fusion of a DJNP with the IPI generated for a series of time points. In this study, an IPI was created every 0.3 s, and all the IPIs were fused with DJNPs.

Figure 7 presents the skewness or displacement for a walking video at three time points (i.e., *t*_0_, *t*_1_, and *t*_2_). Figure 7A,C,D,F depict DJNPs and IPIs for skewed walking. Figure 7B,E depict DJNPs and IPIs for straight walking. These DJNPs can be used to measure skewness and horizontal postural movement.

The parameters *Θ**_r_* and *Θ**_l_* represent the angles of the right and left sides of the body during captured images, respectively (Figure 7E). The ratio of two angles (i.e., *SR* = *Θ**_l_*/*Θ**_r_*) was used to measure the skewness tendency. When this ratio is >1, the body tends to skew to the right. When *SR* = 1, the body is almost straight. When *SR* is <1, the body tends to skew to the left. The displacement of the body between two time points was quantified by estimating the distance covered between these time points. For example, in Figure 7B,E, *D*_*r*,0,1_ and *D*_*l*,0,1_ represent the displacements of the right and left sides of the body, respectively between *t*_0_ and *t*_1_. Similarly, *D*_*r*,1,2_ and D_*l*,1,2_ represent the displacements of the right and left sides of the body, respectively, between *t*_1_ and *t*_2_. Therefore, the ratio of *D*_*r*,*i*-1,*i*_ to *D*_*l*,*i*-1,*i*_ (i.e., *MD* = *D*_*r*,*i*-1,*i*_/*D*_*l*,*i*-1,*i*_, *i* = 0, 1, 2) could be used to determine the dominant side of body displacement. When *MD* was >1, the right side was the dominant side of displacement. When *MD* was 1, the walking posture was almost straight. Moreover, when *MD* was <1, the left side was the dominant displacement side.

### 4.2. Literature for Health Issues and Postural Control during Walking 

Poor postural control during walking may indicate health problems. An individual’s postural control considerably influences their quality of life [2,3]. Equipping participants with wearable devices that assess their posture can be challenging [4]. Nevertheless, this problem can be overcome by incorporating deep learning into Internet of things monitoring systems to effectively detect motion and posture [5]. Resting tremors and finger tapping have been detected using OpenPose-based deep learning methods [6,7]. Moreover, skeleton normality has been determined through the measurement of angles and velocities by using the aforementioned methods [8,9,10]. Such methods are useful for not only generating three-dimensional poses [11,12] but also for identifying the relationship between postural behavior and functional diseases, such as Parkinson’s disease [6,13,14], autism spectrum disorder [15], and metatarsophalangeal joint flexions [16]. OpenPose-based deep learning methods can be used for skeleton, ankle, and foot motion [8,17] detection; physical function assessment [18,19]; and poststroke study [20].

Thus, noninvasive tracking devices play crucial roles in the recording [21], analysis, measurement, and detection of body posture, which may indicate health issues in real time.

## 5. Conclusions

In this study, fusion deep learning was applied to generate DJNPs by using an OpenPose-based method and quantify skewness by using CNNs. The adopted approach successfully incorporates cell phone camera recording, cloud storage, and fusion deep learning for posture estimation and classification. Moreover, the adopted IPI method can be used to perform a quantified analysis of postural control and walking behavior.

The research conducted in the present study can be considered preliminary. We developed the IPI method and attempted a quantified analysis of postural control and walking behavior to identify factors indicative of possible clinical gait disorders. However, at the time of writing, the research is in the preliminary phase and will remain as such until the automated analysis is completed through the IPI method. The highlights of our proposed method include its suitability for use with computer vision for identifying signs of gait problems for clinical application, as well as its replacement of a dynamic joint node plot. In addition, the IPI method is straightforward and allows for real-time monitoring. A video of walking behavior can be conveniently recorded in real-time by using a mobile device. A user can easily remove the background from the video and generate dynamic joint node coordinates through fusion AI methods. The developed IPI method allows for use with computer vision to identify postural characteristics for clinical applications.

Future studies can apply the proposed approach to individuals with health problems to validate this approach.

## Figures and Tables

**Figure 1 biosensors-12-00295-f001:**
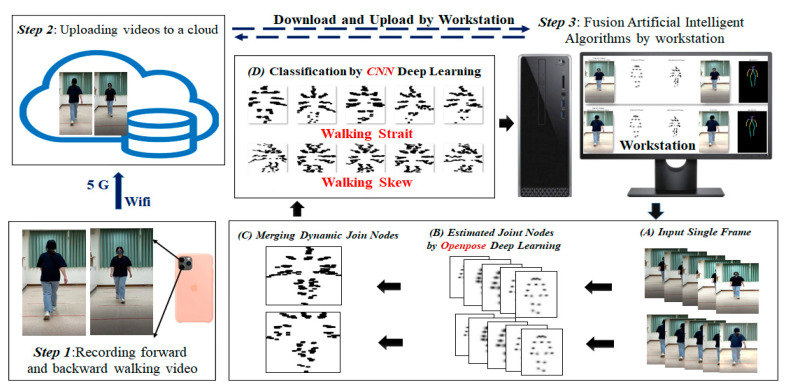
Flow of research.

**Figure 2 biosensors-12-00295-f002:**
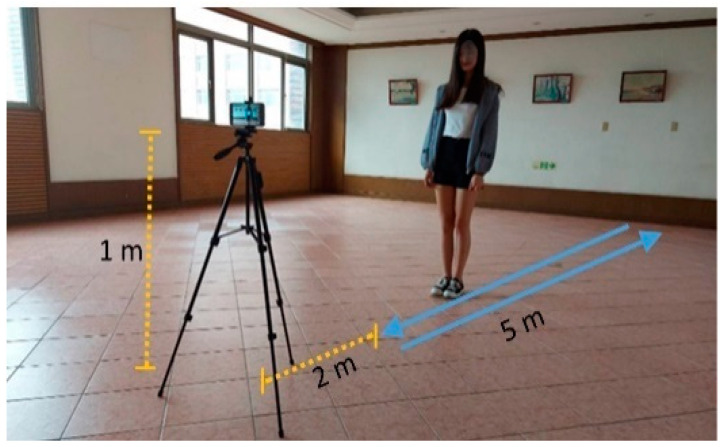
Experimental setup (the cell phone was placed 1 m above the floor and 2 m from the participant).

**Figure 3 biosensors-12-00295-f003:**
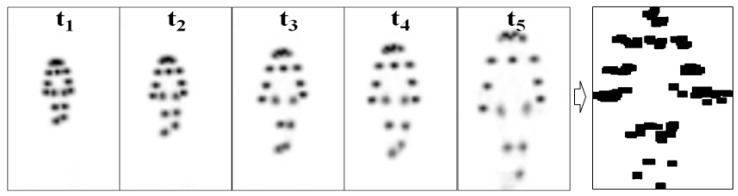
Dynamic joint node plot (DJNP) (right) obtained by merging the heat maps of joint nodes from t_1_ to t_5_ by using the OpenPose algorithm.

**Figure 4 biosensors-12-00295-f004:**
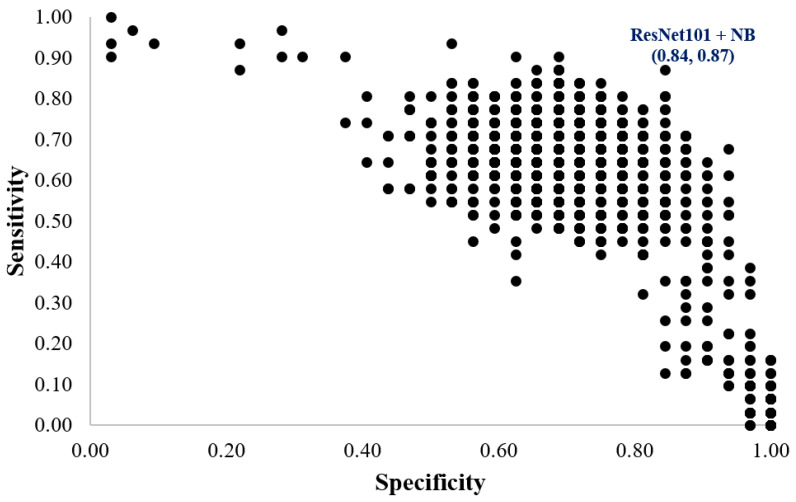
Scatter plot for the specificity and sensitivity of the 1920 models for the validation dataset.

**Figure 5 biosensors-12-00295-f005:**
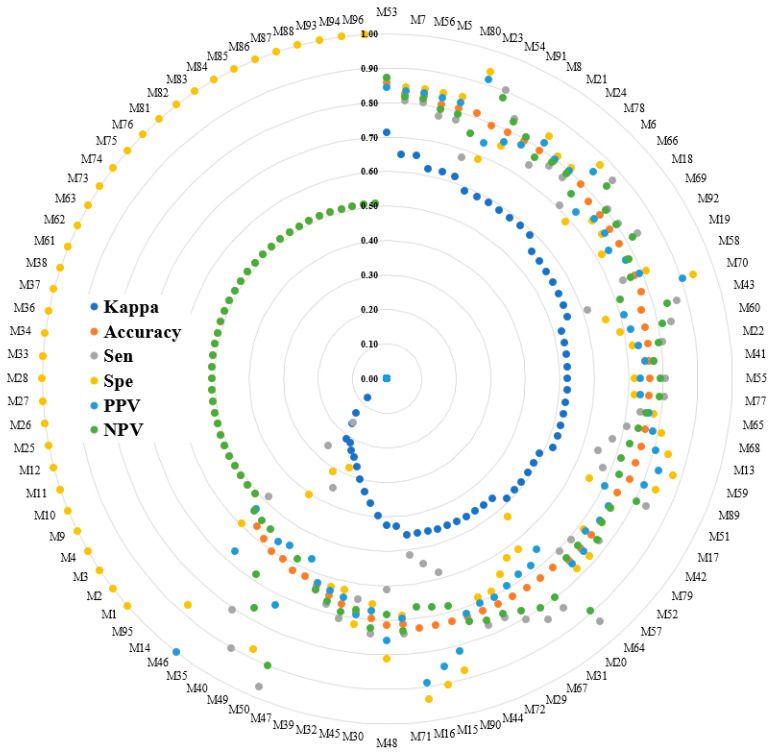
Radar plot of the six performance indices sorted in the ascending order of the kappa value for 96 models (the abbreviations are explained in the Appendix A). Sen represents the sensitivity, and Spe represents the specificity.

**Figure 6 biosensors-12-00295-f006:**
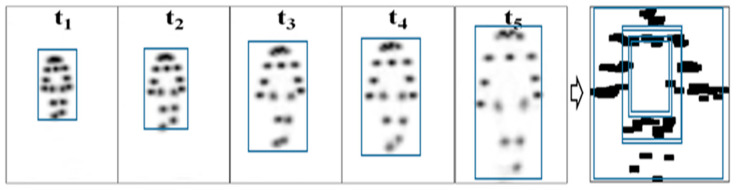
Iso-block postural identity (IPI) generated for a series of times and fusion of the IPI with a DJNP (right).

**Figure 7 biosensors-12-00295-f007:**
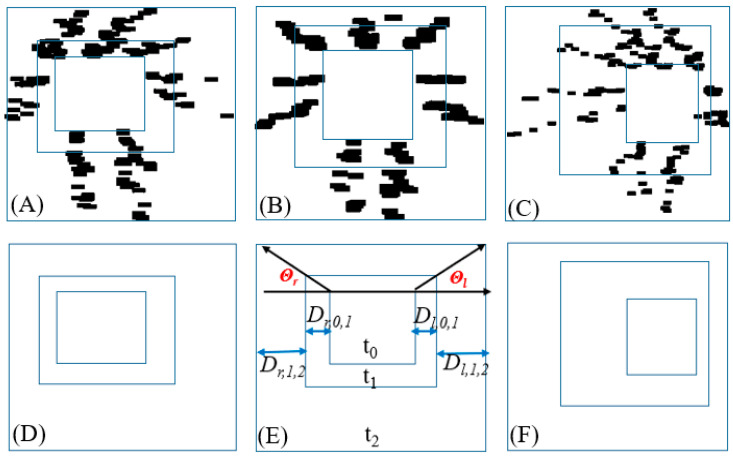
Graphical representation of the skewness or displacement for a walking video at three time points (i.e., *t_0_, t_1_, t_2_*). (**A**,**D**), (**B**,**E**), and (**C**,**F**), respectively, present postural skew to the left, postural balance, and postural skew to the right with participants walking toward the camera.

**Table 1 biosensors-12-00295-t001:** Information on the number of participants and the mean and standard deviation (STD) of velocity (m/s) and time (s) for each group.

Group	N	Mean Velocity (m/s)	STD Velocity (m/s)	Mean Time (s)	STD Time (s)
Skew	102	0.68	0.08	7.48	0.84
Straight	108	0.69	0.08	7.39	0.91

**Table 2 biosensors-12-00295-t002:** Information on the adopted convolutional neural networks.

CNN	Image Size	Layers	Parametric Size (MB)	Layer of Features
AlexNet	227 × 227	25	227	17th (4096 × 9216)
DenseNet201	224 × 224	709	77	706th (1000 × 1920)
GoogleNet	224 × 224	144	27	142nd (1000 × 1024)
MobileNetV2	224 × 224	154	13	152nd (1000 × 1280)
ResNet101	224 × 224	347	167	345th (1000 × 2048)
ResNet50	224 × 224	177	96	175th (1000 × 2048)
VGG16	224 × 224	41	27	33rd (4096 × 25,088)
VGG19	224 × 224	47	535	39th (4096 × 25,088)

**Table 3 biosensors-12-00295-t003:** Models with kappa values greater than 0.59.

CNN	Classifier	Batch Size	Model	Kappa	Accuracy	Sen	Spe	PPV	NPV
ResNet101	NB	5	M53	0.71	0.86	0.87	0.84	0.84	0.87
AlexNet	NB	11	M7	0.65	0.83	0.81	0.84	0.83	0.82
ResNet101	NB	14	M56	0.65	0.83	0.81	0.84	0.83	0.82
AlexNet	NB	5	M5	0.62	0.81	0.77	0.84	0.83	0.79
VGG16	NB	14	M80	0.62	0.81	0.77	0.84	0.83	0.79
DenseNet201	SVM	11	M23	0.62	0.81	0.68	0.94	0.91	0.75
ResNet101	NB	8	M54	0.59	0.79	0.90	0.69	0.74	0.88
VGG19	NB	11	M91	0.59	0.79	0.84	0.75	0.77	0.83
AlexNet	NB	14	M8	0.59	0.79	0.81	0.78	0.78	0.81
DenseNet201	SVM	5	M21	0.59	0.79	0.74	0.84	0.82	0.77
DenseNet201	SVM	14	M24	0.59	0.79	0.77	0.81	0.80	0.79
VGG16	NB	8	M78	0.59	0.79	0.77	0.81	0.80	0.79
AlexNet	NB	8	M6	0.59	0.79	0.71	0.88	0.85	0.76

## Data Availability

Not applicable.

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
