# Peer review of "Identifying the Posture of Young Adults in Walking Videos by Using a Fusion Artificial Intelligent Method"

_biosensors, 2022, doi:10.3390/bios12050295_

Round 1

Reviewer 1 Report

I consider of interest investigation regarding the development of noninvasive tracking methodologies. The paper focus on that interested on analyzing the posture of young adults when walking. I consider that the main goal could be better defined, some more details are interesting in the methods section and is in the discussion that in my opinion the authors could improve considerably starting from the defined goal(s).

Specific comments below:

Methods section

  • Do not find any information regarding the walking intensity/velocity that the participants were instructed to perform.
  • 6. Present the number of participants that were allocated to each group – experimental and control.

Results section

  • L193 L195 – naïve instead of Naïve.

Discussion section

  • 1 subsection continues to be results presentation. Do not see discussion in these presentations of figures 6 and 7.
  • Figure 7 title. It should be introduced on the end ”, respectively”.
  • L241/242 Considering that during the paper the use of wearable devices for postural analysis is not mentioned, considered of interest to develop this “comparison” in depth.

Discussion should be improved considerably analysing and discussion the control and the experimental groups, the results in terms of the different options in terms of the fusion of artificial intelligent methods as well as contribution to health issues and postural control during walking.

Author Response

I consider of interest investigation regarding the development of noninvasive tracking methodologies. The paper focus on that interested on analyzing the posture of young adults when walking. I consider that the main goal could be better defined, some more details are interesting in the methods section and is in the discussion that in my opinion the authors could improve considerably starting from the defined goal(s).

Specific comments below:

Methods section

Do not find any information regarding the walking intensity/velocity that the participants were instructed to perform.

Present the number of participants that were allocated to each group – experimental and control.

[Reply]: Thank you for the comment. The relevant numbers, velocity (m/s), and walking time have been added to Table 1 in this revision.

Table 1 lists the number of participants and the mean and standard deviation (STD) of velocity (m/s) and time (s) for each group.”

Results section

L193 L195 – naïve instead of Naïve.

[Reply]: Thank you for the comment. We have replaced “Naïve” with “naïve.”

Discussion section

1 subsection continues to be results presentation. Do not see discussion in these presentations of figures 6 and 7.

[Reply]: We have updated Section 4.1:

“Figure 6 illustrates the fusion of a DJNP with the IPI generated for a series of time points. In the present study, an IPI was created every 0.3 s, and all the IPIs were fused with DJNPs.
Figure 7 plots the skewness or displacement for a walking video at three time points (i.e., t0, t1, and t2). Figure 7A, 7C, 7D, and 7F depict the DJNPs and IPIs for skewed walking. Figure 7B and 7E depict the DJNPs and IPIs for straight walking. These DJNPs can be used to measure skewness and horizontal postural movement.”

Figure 7 title. It should be introduced on the end ”, respectively”.

[Reply]: Thank you for the comment. In this revision, we have modified the title as follows: “Graphical representation of the skewness or displacement for a walking video at three time points (i.e., t0, t1, t2). (A, D), (B, E), and (C, F) respectively present postural skew to the left, postural balance, and postural skew to the right with participants walking toward the camera.

L241/242 Considering that during the paper the use of wearable devices for postural analysis is not mentioned, considered of interest to develop this “comparison” in depth.

[Reply]: In accordance with your suggestion, references pertaining to the use of wearable devices for postural analysis have been added to the revised manuscript.

Poor postural control during walking may indicate health problems. An individual’s postural control considerably influences their quality of life [2,3]. Equipping participants with wearable devices that assess their posture can be challenging [4]. Nevertheless, this problem can be overcome by incorporating deep learning into Internet of things monitoring systems to effectively detect motion and posture [5].”

In the near future, an in-depth comparison of studies exploring the use of wearable devices for postural analysis should be conducted.

Discussion should be improved considerably analysing and discussion the control and the experimental groups, the results in terms of the different options in terms of the fusion of artificial intelligent methods as well as contribution to health issues and postural control during walking.

[Reply]: We have added the highlights of our proposed method to the Conclusion section of the revised manuscript.

We developed the IPI method and attempted a quantified analysis of postural control and walking behavior to identify factors indicative of possible clinical gait disorders. However, at the time of writing, the research is in the preliminary phase and will remain as such until the automated analysis is completed through the IPI method. The highlights of our proposed method include its suitability for use with computer vision for identifying signs of gait problems for clinical application as well as its replacement of a dynamic joint node plot. In addition, the IPI method is straightforward and allows for real-time monitoring. A video of walking behavior can be conveniently recorded using a mobile device. A user can easily remove the background from the video and generate dynamic joint node coordinates through fusion AI methods.”

Reviewer 2 Report

The authors describe their experiments in walking and discrimination between straight and skewed walking. The colected videos were analyzed using many types of neural networks. In principle, the manuscript is well written. However, there are some points that need improvement or clarification.

  1. The aim - classification of walking types should be described in more detail, including characteristics of the different walking types.
  2. Subsection 2.6: Was the walking assessed by a medical expert?
  3. Subsection 2.7: there is a list of NNs and other classifiers. There is no reference at any of them.
  4. Subsection 2.7: it is not clear which features serve as input for LR, NB and SVM.
  5. Section 3: the division of data into training and validation sets is not clear. Were samples of the same person used either for training or validation only or were these samples in both sets? Please explain division of data in detail.
  6. Section 3, line 185: It is not clear what 1920 models represent. Please explain in detail.

Author Response

The authors describe their experiments in walking and discrimination between straight and skewed walking. The collected videos were analyzed using many types of neural networks. In principle, the manuscript is well written. However, there are some points that need improvement or clarification.

The aim - classification of walking types should be described in more detail, including characteristics of the different walking types.

Subsection 2.6: Was the walking assessed by a medical expert?

[Reply]: Our walking videos featured only healthy young adults. Furthermore, the walking videos were not assessed by a medical expert.

Subsection 2.7: there is a list of NNs and other classifiers. There is no reference at any of them.

[Reply]: References related to CNNs and classifiers have been added to the revised manuscript.

“CNNs have demonstrated utility and efficiency in image feature extraction in the fields of biomedicine and biology [2327].”

Subsection 2.7: it is not clear which features serve as input for LR, NB and SVM.

[Reply]: In the revised manuscript, “Table 2 lists the features that were extracted by CNNs and served as the inputs for the LR, NB, and SVM.

Section 3: the division of data into training and validation sets is not clear. Were samples of the same person used either for training or validation only or were these samples in both sets? Please explain division of data in detail.

[Reply]: “The data for the control and experimental groups comprised 102 and 108 DJNPs, respectively. A random splitting schema was employed to separate the training (70%) and testing (30%) sets; 71 and 31 samples from the control group were used for training and testing, respectively, and 76 and 32 samples from the experimental group were used for training and testing, respectively.

This sentence has been added to the revised manuscript.

Section 3, line 185: It is not clear what 1920 models represent. Please explain in detail.

[Reply]: “In the present study, eight CNNs and three classifiers with four batch sizes and 20 random splits were adopted. Therefore, the 1,920 models represent the 1,920 possible combinations of one CNN, classifier, batch size, and random data split.”

This sentence has been added to the revised manuscript.

Reviewer 3 Report

In this study, fusion deep learning was applied to generate DJNPs using an Open-Pose-based method and quantify skewness using CNNs. The method and the AI technological tools used in the research undoubtedly represent topics of major interest in health monitoring fields.

Despite the adopted approach demonstrated to be successful in incorporating cell phone camera recording, cloud storage, and fusion deep learning for posture estimation, the kind of classification presented is elementary. Indeed, the adopted IPI method is not sufficient to perform a quantified analysis of postural control and walking behaviour in any possible health/clinical gait disorders application as claimed in the conclusions. Thus, the research can be considered at a preliminary stage. The authors are warmly encouraged to use their method to focus on the quantification/identification/classification of specific gait disorders characteristics and compare their results with other validated technology for quantitative gait and posture analysis to validate it and highlight the proposed method's peculiarities.

Author Response

In this study, fusion deep learning was applied to generate DJNPs using an Open-Pose-based method and quantify skewness using CNNs. The method and the AI technological tools used in the research undoubtedly represent topics of major interest in health monitoring fields.

Despite the adopted approach demonstrated to be successful in incorporating cell phone camera recording, cloud storage, and fusion deep learning for posture estimation, the kind of classification presented is elementary. Indeed, the adopted IPI method is not sufficient to perform a quantified analysis of postural control and walking behaviour in any possible health/clinical gait disorders application as claimed in the conclusions. Thus, the research can be considered at a preliminary stage. The authors are warmly encouraged to use their method to focus on the quantification/identification/classification of specific gait disorders characteristics and compare their results with other validated technology for quantitative gait and posture analysis to validate it and highlight the proposed method's peculiarities.

[Reply]: Yes. “The research conducted in the present study can be considered preliminary. We developed the IPI method and attempted a quantified analysis of postural control and walking behavior to identify factors indicative of possible clinical gait disorders. However, at the time of writing, the research is in the preliminary phase and will remain as such until the automated analysis is completed through the IPI method. The highlights of our proposed method include its suitability for use with computer vision for identifying signs of gait problems for clinical application as well as its replacement of a dynamic joint node plot. In addition, the IPI method is straightforward and allows for real-time monitoring. A video of walking behavior can be conveniently recorded in real-time by using a mobile device. A user can easily remove the background from the video and generate dynamic joint node coordinates through fusion AI methods. The developed IPI method allows for use with computer vision to identify postural characteristics for clinical applications.

This content has been added to the revised manuscript.

Round 2

Reviewer 1 Report

No further suggestions.

Author Response

Thank you for your valuable time and comments that have helped us a lot.

Reviewer 2 Report

The manuscript has been improved and some points have been clarified. However, there are still some issues. The questions and comments were reflected in the authors´ response but not so much in the text.

Subsection 2.6: Was the walking assessed by a medical expert?

[Reply]: Our walking videos featured only healthy young adults. Furthermore, the walking videos were not assessed by a medical expert.

Please add to the text of the manuscript who assessed the videos.

Subsection 2.7: it is not clear which features serve as input for LR, NB and SVM.

[Reply]: In the revised manuscript, “Table 2 lists the features that were extracted by CNNs and served as the inputs for the LR, NB, and SVM.

The table 2 needs clarification. There is a column Layer of features. However, there is no information about the features as such. So it remains still unclear what the input to the classifiers is.

Section 3: the division of data into training and validation sets is not clear. Were samples of the same person used either for training or validation only or were these samples in both sets? Please explain division of data in detail.

[Reply]: “The data for the control and experimental groups comprised 102 and 108 DJNPs, respectively. A random splitting schema was employed to separate the training (70%) and testing (30%) sets; 71 and 31 samples from the control group were used for training and testing, respectively, and 76 and 32 samples from the experimental group were used for training and testing, respectively.

This sentence has been added to the revised manuscript.

This is clear now. Thank you.

I have here one additional question: How many static frames are in one DJNP?

Please check mistyping errors.

Author Response

We thank Reviewer 2 for the useful suggestions. We changed the manuscript accordingly. 

Reviewer 3 Report

Being at the preliminary stage, as admitted by the authors, the research is not at the level to be published. I suggest the authors take into account the suggestions given in the round #1 revision. The paper in the present stage is not admittable to be published and thus is to be rejected.

Author Response

We are deeply sorry for this decision.
The research is in the preliminary phase and will remain as until the automated analysis is completed  for identifying signs of gait problems for clinical application.
Thank you for your valuable time and comments.